# Implementation of Virtual Antenatal and Postnatal Urgent Midwifery Visits: Evaluation of a Quality Improvement Initiative

**DOI:** 10.3390/ijerph21070903

**Published:** 2024-07-10

**Authors:** Nora Drummond, Joanne Bailey, Christina Majszak, Ruth Zielinski

**Affiliations:** 1School of Nursing, University of Michigan, Ann Arbor, MI 48109, USA; ruthcnm@umich.edu; 2University of Michigan Health—Midwifery Service, University of Michigan, Ann Arbor, MI 48109, USA; jabailey@umich.edu (J.B.); cmmajsza@umich.edu (C.M.)

**Keywords:** antenatal, postnatal, urgent care, virtual visits, telehealth, midwifery

## Abstract

Women seeking care during the perinatal period often face delays or long waits at healthcare facilities due to lack of providers and/or resources, leading to sub-optimal outcomes. We implemented a program whereby patients with concerns could receive same-day care virtually from a midwife rather than presenting to the clinic or hospital for care. Implementation strategies included virtual training, a staged increase in patient volume, and frequent communication between the midwives via text, email, and monthly meetings. Virtual visits included a variety of complaints, the five most common being to establish care, first-trimester bleeding, nausea and vomiting, mental health concerns, and postnatal breast problems. There was a threefold increase in virtual visits during the first 6 months with 92% of patients not requiring urgent face-to-face follow-up. Midwives were able to provide high-quality telehealth care that met the patients’ needs and decreased the demand on hospital-based services. With the growing ubiquity of mobile phones and internet access, this strategy may be effective in providing quality care while decreasing demands on physical infrastructure. More research is needed to assess acceptability in other contexts. Reproducibility in low-resource settings may be limited if women lack access to video conferencing on phones or laptops.

## 1. Introduction

For optimal maternal and neonatal outcomes, the World Health Organization (WHO) recommends that women receive at least eight visits during the antenatal period and at least three visits during the postnatal period [1]. According to the most recent data, while 88% of women throughout the world receive at least one visit, only 69% receive even four antenatal visits [2]. Prior to the COVID-19 pandemic, maternity care facilities were already challenged by the influx of patients seeking care during the antenatal period. This increased demand may be fueled in low-resource settings by underdevelopment of the maternity care workforce and infrastructure or in high-resource settings by the decline and closure of available maternity care hospitals [2]. During the COVID-19 pandemic, the number of antenatal and postnatal visits decreased, due in part to overwhelmed healthcare systems [3]. As a result, there is evidence of an increase in stillbirth, maternal death, ectopic pregnancy, and maternal depression globally [4].

Telehealth, or virtual healthcare via phone or video conferencing platforms, has shown utility for a variety of healthcare services. Virtual routine antenatal visits have been implemented to a very limited extent as an approach to meet patient needs. When interspersed with in-person visits, patients were satisfied with the convenience of virtual visits [5]. Virtual visits have also been used successfully for consultation with healthcare specialists such as maternal–fetal medicine and lactation support [6]. However, utilization for antenatal and postnatal care remains low with an estimated 0.1% of all services delivered virtually [6].

While routine antenatal and postnatal care is important for maternal and infant well-being, questions and concerns often arise between those visits. Women may wait until their next scheduled visit, which could result in a delay in care and potential maternal and fetal complications [7]. When women present to maternity care facilities, they are “triaged” or prioritized in order to attend to the most critical patients first, which can result in long wait times and low patient satisfaction [8]. Obstetric triage visits contribute to a large volume of patient visits, exceeding the overall birth volume in facilities by 20–50% [9].

In our institution, patients with urgent questions or needs were instructed to call a phone nurse who could direct the patient to present to the hospital triage unit for same-day care. This included patients with both emergent and urgent, non-emergent issues. This resulted in a triage department that was often over-full and long patient wait times. In an effort to provide timely care and decrease patient care burden in a large maternity care center, we implemented a program of virtual urgent visits staffed by certified nurse-midwives (CNMs). Our aim in this article is to describe the implementation and evaluation of a program of virtual visits to improve the quality of peripartum care given at our institution.

## 2. Materials and Methods

Implementation and evaluation of the virtual CNM urgent visit program was informed by the Ottawa Model of Research Use (OMRU) [10]. This quality improvement model has been utilized in initiatives across practice specialties and patient lifespans [11,12,13,14]. Recently, this model has been operationalized in settings relevant to the virtual urgent visit program including quality improvement initiatives on labor and delivery units and in evaluations of telehealth programs [15,16].

For this project, the six constructs of the Ottawa Model of Research Use (evidence-based innovation, potential adopters, practice environment, implementation strategies adoption, and outcome) were operationalized into implementation and evaluation processes. The project was reviewed by the Institutional Review Board at the University of Michigan and determined to be non-regulated (HUM00237639). Chart review and a survey of the midwives completed at one month and one year were reviewed. The initial survey focused on knowledge and beliefs about the new role and included 19 questions of various types including short answer, multiple choice, and questions measured on a 5-point scale from “strongly disagree to strongly agree”. The follow-up survey consisted of 12 questions, utilizing the same variety in question type, and was focused on how the midwives’ knowledge and beliefs about the role had changed over the course of the year. The quantitative data generated from the 5-point scale and multiple-choice questions were enhanced and contextualized by the comments provided by the midwives in the survey. Descriptive statistics were utilized for the evaluation of the program and categorically organized using the constructs of the OMRU. The implementation and evaluation strategies are summarized in Figure 1.

## 3. Results

### 3.1. Implementation

#### 3.1.1. Practice Environment

Central to the virtual urgent visit program was a shift in the practice environment from an in-person urgent visit within a medical center to the homes of the midwives themselves. Patients were provided with 24 h phone access to a triage nurse who assessed the need for an urgent visit. The program offered virtual urgent visits available between 7 a.m. and 7 p.m., 7 days/week. Patients calling with concerns outside of that time frame were directed to the hospital for inpatient triage care. Additionally, when patient concerns necessitated immediate in-person evaluation, such as working up preeclampsia, the midwives were able to direct the patient to the inpatient setting for care. The same midwives working in this role were also managing a postpartum hypertensive disorder text-based surveillance program. Virtual urgent care visits were conducted via a private secure link from the electronic medical record patient portal with Zoom^®^ functionality. Midwives received hardware including a laptop computer equipped with a built-in camera and monitor and software including charting applications accessed through secure internet portals to protect patient privacy.

#### 3.1.2. Potential Adopters

The potential adopters of this intervention were CNMs already working within the hospital system. All midwives trained for the program had previously worked in the obstetric triage unit within the hospital. The midwives were offered optional computerized training modules in virtual care, a practice virtual session, and shadowing of other providers performing virtual visits. Most of the midwives (6/9) completed the virtual training and participated in a practice virtual visit, and one-third of the midwives (3/9) shadowed another provider performing virtual visits. Of note, only one midwife elected to complete the ambulatory basics training, which covered outpatient charting. The outpatient setting was new to all midwives except one who had previously worked in an ambulatory clinic. This was a missed opportunity as many of the midwives felt challenged by utilizing the electronic health record and documenting their visits in the ambulatory setting.

#### 3.1.3. Intervention Strategies

Creating mechanisms for patients to schedule urgent visits was central to the success of the intervention. The staff involved in routine scheduling and the nurses working in the ambulatory care sites participated in presentations of this new role. Regular feedback was solicited about barriers to scheduling patients and clarifying appropriate patients for the virtual visits. Informal messaging between virtual midwives, schedulers, and nurses managing patient phone calls occurred daily to appropriately schedule patients for virtual urgent visits, routine care, or emergency triage visits.

#### 3.1.4. Adoption

A plan was made for a staged roll-out of the program with gradual increases in patient volume over time. The midwives were supported formally with bi-weekly staff meetings to address any protocol changes or patient challenges and informally with midwife-led text and email chains. Standard note templates for the most common conditions were developed and implemented.

#### 3.1.5. Outcomes

A plan was made to evaluate the response of patients to the program. Chart review was utilized to assess the availability of visits, no-show rates, and services the midwives provided. As direct measures of patient satisfaction were unavailable, continuation of care within the healthcare system after an urgent virtual visit to establish care was utilized as a proxy for patient satisfaction, as patients have many options for their perinatal care in this geographic location.

### 3.2. Evaluation

#### 3.2.1. Practice Environment

The transition to the digital environment was complicated by one midwife having repeated difficulty accessing equipment and another needing a new camera for her system at home. The midwives overall were satisfied with their technological support services, with eight of nine midwives either somewhat or strongly agreeing that they had easy access to technological support when needed. By one year into the program, the early issues with hardware were resolved, with all midwives somewhat agreeing or strongly agreeing that they had adequate at-home equipment and technical support to do their work.

#### 3.2.2. Potential Adopters

At one month of caring for patients via virtual urgent visits, the midwives felt that the training could have been more robust, with nearly half somewhat disagreeing that they had adequate training for their new role (Table 1). The midwives were split on their satisfaction with the new role, with three midwives somewhat or strongly agreeing that they were enjoying the new role and the rest either neutral or somewhat disagreeing. Some midwives experienced stress adapting to the online environment where they did not have instant access to a robust physical exam, vital signs, lab values, or imaging studies. The midwives also had to train in a new charting system, which proved challenging. Most of the midwives had not worked in the ambulatory care setting recently, so the systems for scheduling patients and navigating diverse clinic processes were new to them. The midwives as a team had to work together to create follow-up processes for labs and imaging ordered by one midwife when the results were not available until a different midwife was on duty. One midwife summed up the initial difficulties: “The learning curve was much steeper than anticipated for most of us…those who do not work outpatient need more orientation to that role, how to communicate with other providers, referral processes, charting in the outpatient setting”. By one year of practice in the virtual role, the midwives had adapted well to the job, with all the midwives rating their experience working in the role as good or great. One midwife summarized her satisfaction in the role: “I can spend significantly more time advocating, teaching, and supporting. And that is why I became a midwife”.

#### 3.2.3. Intervention Strategies

Scheduling patients with the midwives evolved over time. In the first two months visits were limited to 2–3 visits per day. This allowed the midwives the time and flexibility to address challenges with technology, patient follow-up, or other issues. After the first two months, visits were scheduled in 30-min blocks and the midwife was given at least 30 min of notice prior to the visit to review the patient’s history and prepare for the visit. Four months into the program, to increase efficiency and consistency throughout the day, patients were scheduled in blocks such as 7–8 a.m., 12–2 p.m., and 6–7 p.m. This allowed midwives to gain efficiency and consistency in their schedules while still offering same-day appointments for patients.

### 3.3. Adoption

Over the course of the first six months, the volume of visits grew significantly, increasing from 38 visits in the first month to 123 visits in the fifth month. The midwives saw a wide variety of patient complaints across the antenatal and postnatal periods (Table 2). The five most common urgent issues were to establish antenatal care, first-trimester bleeding, nausea and vomiting, postnatal breast concerns, and postnatal mood concerns. The midwives provided care including counseling, medication management, diagnostic orders, and referrals.

### 3.4. Outcomes

The virtual urgent visit program was successful in providing timely care to patients while reducing visits to the hospital and urgent office visits. Over the course of six months, 468 patients were scheduled for an urgent visit. Almost all visits, except for establishing antenatal care visits, were scheduled for the same day. The overall no-show rate was just under 8% (37 visits) for the program over the first six months. No follow-up was needed following the virtual visit for 126 patients’ complaints (26%), and 306 patients (65%) had their issues addressed by the midwife with follow-up during their routine scheduled antenatal or postnatal care visits. Only 6% of patients (30) needed to follow-up in hospital triage, and only 2% (11) needed an urgent office visit. In summary, the midwives were able to address 92% of patient concerns without additional burden to the healthcare system, saving approximately 430 in-person visits in the first six months of the program.

To extrapolate patient satisfaction, we assessed the number of patients who continued care with the healthcare system after their urgent care visit. Only 3% (five patients) of these patients transferred their care to an outside healthcare system. Three of the transfers moved to healthcare systems outside the geographic area and two to another local healthcare system.

## 4. Discussion

The evaluation of the virtual care implementation shows promising results for both patient care and midwife satisfaction. Though there were some initial challenges with equipment provision and ambulatory training for the midwives, within a year, the midwives felt comfortable and satisfied in their virtual role. The midwives were adept at meeting patients’ needs while reducing the burden of in-person urgent visits on the healthcare system. The success of this program demonstrates that skilled midwives can provide telehealth access to the right level of care at the right time for patients. This model may be replicated to address issues of access and overcrowding in healthcare systems in high-resource settings and may have utility in low-resource settings as access to technology increases.

The majority of the virtual urgent care visits were in the first trimester of pregnancy or the postnatal period, underscoring the limitations of current care to address the urgent, non-emergent needs of women during the perinatal period. In the first trimester, an “establishing antenatal care” virtual visit acts as a bridge to care prior to a standard new patient visit that is often scheduled beyond the first trimester due to clinic capacity. Expanding the capacity to serve women in this period is vital as first-trimester complications can have deleterious effects on fetal and maternal outcomes [1]. Nausea and vomiting of pregnancy are common complaints that can lead to dehydration and require emergency services if not addressed in the early stages. In the postnatal period, most patients only receive one visit 4–6 weeks after birth. This creates insufficient opportunity to address urgent issues such as lactation difficulties and mood disorders, as well as areas of opportunities for patient education such as sleep, nutrition, and family planning. The establishment of urgent visits can fill this gap in care.

In the United States, reproductive healthcare access is unevenly distributed, with over one-third of counties qualifying as maternity care deserts lacking access to comprehensive reproductive healthcare [17]. Telehealth has been proposed as a potential approach for providing care to those living in these areas [18]. Studies have shown non-inferiority for telehealth replacing some routine antenatal and postnatal care on patient satisfaction and some obstetric outcomes, but higher rates of emergent issues such as preeclampsia and preterm birth have also been noted [17,19]. The virtual urgent visit program was not intended to replace routine in-person antenatal or postnatal visits but rather to provide additional access to care during the perinatal period. By providing same-day access to a maternity care provider, patients were able to discuss worrisome or vague symptoms earlier.

### Limitations

No direct measures of patient satisfaction with virtual urgent visits were collected for this project. This project was implemented in a high-resource area with midwives who were interested in providing virtual visits and may not be transferrable to other contexts. For example, there are components of virtual visits that may be difficult to achieve in low-resource settings. The initial capital investment for computer equipment and programs may be a barrier. The virtual visit service requires reliable cellular or internet services, which may be challenging in rural, underdeveloped, or politically unstable areas. However, access to technology has increased significantly in low- and middle-income countries, and the feasibility of implementing virtual urgent visits with midwives in these settings should be considered.

## 5. Conclusions

Access to quality maternity care is essential for optimizing outcomes for mothers and newborns. The midwife-led virtual urgent visit program shows promise in addressing urgent needs for pregnant and postnatal patients while reducing the stress on the healthcare system. More research is needed to establish the feasibility and efficacy of this model in low-resource settings.

## Figures and Tables

**Figure 1 ijerph-21-00903-f001:**
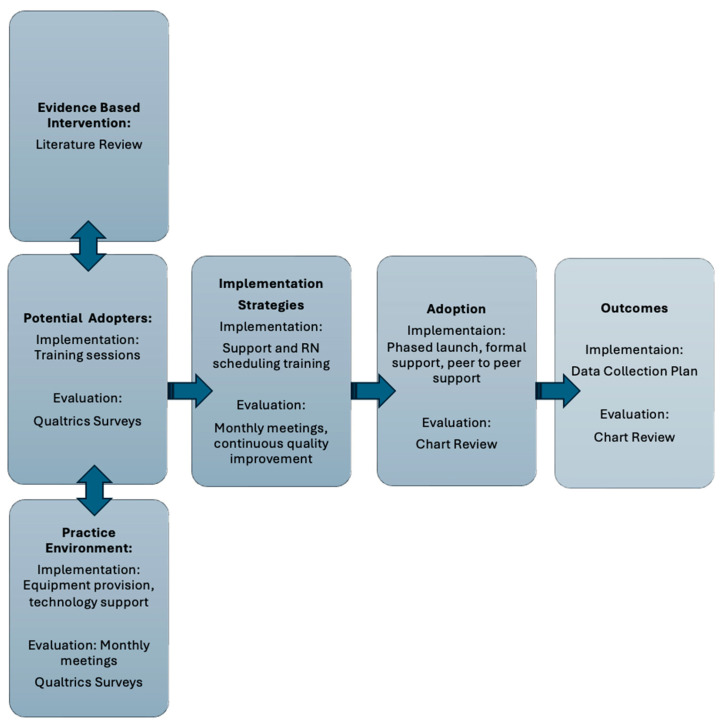
Implementation and Evaluation Plan utilizing the Ottawa Model of Research Use.

**Table 1 ijerph-21-00903-t001:** Virtual midwife survey results.

One Month Survey (N = 7)	Strongly Agree	Agree	Neutral	Disagree	Strongly Disagree	Mean
I have a clear understanding of my role	1	5	0	1	0	3.6
I had adequate training for my role	2	1	1	3	0	3.3
I have the equipment I need	5	1	0	0	1	4.3
Tech support is easily available	4	2	1	0	0	4.1
**Twelve Month Survey (N = 7)**	**Strongly Agree**	**Agree**	**Neutral**	**Disagree**	**Strongly Disagree**	**Mean**
I have a clear understanding of my role	5	2	0	0	0	4.7
I enjoy working from home	5	2	0	0	0	4.7
This has improved my work schedule	6	1	0	0	0	4.9
I have the equipment I need	5	2	0	0	0	4.7
Tech support is easily available	4	1	2	0	0	4.3
I enjoy my role as a virtual midwife	5	2	0	0	0	4.7

**Table 2 ijerph-21-00903-t002:** Virtual Visit Patient Complaints.

Chief Complaint:	April	May	June	July	August	September	Total	%
Establish Care	1	0	20	41	67	52	181	39
First-Trimester Bleeding	7	10	14	10	10	11	62	13
Breast Complaints	5	3	10	14	12	9	53	11
Nausea and Vomiting	0	7	7	11	14	5	44	9
No-show	6	3	7	2	7	10	35	7
Urinary Complaint	4	1	5	0	0	0	10	2
Other *	15	11	18	15	13	11	83	18
Total	38	35	81	93	123	98	468	100

* Other includes a total of four or fewer visits for each of the following complaints: abdominal pain, dizziness, pelvic pain, antepartum mood complaints, preterm contractions, vulvar irritation, ectopic pregnancy, deep vein thrombosis, leg pain, thyroid issues, cellulitis, fatigue, gestational diabetes, headache, lupus, postpartum laceration, postpartum bleeding, incision pain, decreased fetal movement, medication questions, postpartum fecal incontinence, carpal tunnel, and postpartum grief.

## Data Availability

Data are available upon request to the corresponding author Nora Drummond noradrum@umich.edu.

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
