# Peer review of "Implementation of Virtual Antenatal and Postnatal Urgent Midwifery Visits: Evaluation of a Quality Improvement Initiative"

_ijerph, 2024, doi:10.3390/ijerph21070903_

Round 1

Reviewer 1 Report

Comments and Suggestions for Authors

Dear Authors

There are some points which are better to be considered,

1-Since some examinations are necessary to rule out an urgency or emergency in obstetrics  (for example measurement of blood pressure in severe signs of preeclampsia such as headache or blurred vision) how can you be sure to diagnose an emergency properly?

2-There is no quantitative measurement of midwives' satisfaction in your manuscript.

3-As in obstetrics, there are plenty number of emergency during 24 hours how can you be sure of covering all of them when patients have a complaint about extra of your scheduled time mentioned in line178?

4-Please explain exactly the route of the relationship between pregnant women with the midwifery system. 

Author Response

Reviewer 1

Author response

Since some examinations are necessary to rule out an urgency or emergency in obstetrics  (for example measurement of blood pressure in severe signs of preeclampsia such as headache or blurred vision) how can you be sure to diagnose an emergency properly?

This is an important point – the virtual midwives were able to refer patients needing in-person care to the facility to be seen urgently – we have added clarification of this – Line 105

There is no quantitative measurement of midwives' satisfaction in your manuscript.

The survey data was included in part in the text of the paper – we have added a table with select items from the 1 month and 12 month survey – Line 178

As in obstetrics, there are plenty number of emergency during 24 hours how can you be sure of covering all of them when patients have a complaint about extra of your scheduled time mentioned in line 178

We have added text to clarify that patients have access to phone nurses and all urgent/emergent needs are met in the facility setting when virtual midwives are not on duty – Line 104

Please explain exactly the route of the relationship between pregnant women with the midwifery system. 

Patients have 24 hour access to a triage nurse who determines appropriateness of a virtual urgent visit. This has been clarified in the manuscript - Line 102

Reviewer 2 Report

Comments and Suggestions for Authors

Dear authors; I like the perspective of your article which offers innovative approach alternatives to the antenatal problems we will face in the future. Congratulations. In the future, the importance of technological virtual visits will increase day by day due to climate crisis and wars. I am sending you the comments that will strengthen your article in a pdf file. After these comments, the impact of your article will increase. I wish you continued success.

Author Response

Reviewer 2

Author response

Please add an example of this sentence to the end of the Introduction ; ‘ Our aim in this article is to seek answers to the question of whether we can create a new perspective to prevent maternal fetal complications with the developing technology

A comparable sentence was added at the location the reviewer suggested – Line 59

The first sentence of the discussion title should be the most crucial sentence of the results of the article. For this reason, it would be better to add a sentence similar to the following sentence; ' According to our virtual visit results, we have shown that our results with both midwives and patients are promising and that there is an area open to development.

A comparable sentence was added at the location the reviewer suggested – Line 224

Please consider to add ;’ The impact of virtual visits in reducing maternal fetal complications is undeniable, especially in socio-culturally low societies where access to hospitals or health professionals is difficult.’

Thank you for this suggestion. While we do hope that virtual visits have this impact, the data is not strong enough at this time, more research is needed. We did not want to overstate our results and hope that future research will determine the impact of virtual visits

Please add this sentence and give reference;' Symptoms experienced during first trimester pregnancy have very complicated effects on both fetal and maternal health.

A sentence was added – Line 240

Please consider adding this sentence and give attribution; In the world of the future, healthy nutrition will be one of the most important issues during pregnancy. From this point of view, it is very clear that one of the benefits of virtual visits will be to shed light on the healthy nutritional status of the expectant mother and its possible fetal effects and will be protective against scenarios such as neonatal birth pain, preterm birth or low birth weight.

A sentence was added- Line 246